# The Health Education and Training (HEAT) Corps: A Medical–Community Collaboration Response during the COVID-19 Pandemic

**DOI:** 10.3390/ijerph20031829

**Published:** 2023-01-19

**Authors:** Panagis Galiatsatos, Vanya Jones, Jacqueline Bryan, Meghan Brown, Olivia Banks, Brittany Martin, Megan Collins, Catherine Ling, Mindi B. Levin, Audrey Johnson, Alicia Wilson, Annette Anderson

**Affiliations:** 1Division of Pulmonary and Critical Care Medicine, Department of Medicine, The Johns Hopkins University School of Medicine, Baltimore, MD 21205, USA; 2Medicine for the Greater Good, The Johns Hopkins University School of Medicine, Baltimore, MD 21205, USA; 3Office of Diversity, Inclusion, and Health Equity, The Johns Hopkins University School of Medicine, Baltimore, MD 21205, USA; 4Health, Behavior and Society, Johns Hopkins Bloomberg School of Public Health, Baltimore, MD 21205, USA; 5Office for Economic Development and Community Partnerships, Johns Hopkins University and Johns Hopkins Health System, Baltimore, MD 21205, USA; 6Wilmer Eye Institute, The Johns Hopkins University School of Medicine, Baltimore, MD 21205, USA; 7Johns Hopkins School of Nursing, Baltimore, MD 21205, USA; 8SOURCE, The Community Engagement and Service-Learning Center, Johns Hopkins University Schools of Public Health, Nursing, and Medicine, Baltimore, MD 21205, USA; 9Johns Hopkins School of Education, Baltimore, MD 21205, USA

**Keywords:** COVID-19, community engagement, health equity, curriculum

## Abstract

With the declaration of the COVID-19 pandemic by the World Health Organization in March 2020, many elements of society were faced with attempting to assimilate public health recommendations for infectious control. Vital social organizations had to balance delivering their social services while attempting to stay up to date with COVID-19 information and comply with evolving regulations. In the realm of schools and school systems, guidance on how to best adapt to COVID-19 was often limited. School officials and staff had to assist with multiple public health crises as a consequence of the pandemic, from the pandemic’s transmission prevention strategies (e.g., face masks and physical distancing) to the recognition that students would have personal tragedies related to COVID-19. In this review, we highlight the process and feasibility of implementing an international COVID-19 school-based initiative over two years of the pandemic, the Health Education and Training (HEAT) Corps program.

## 1. Introduction

With the global health crisis brought on by the contagious Severe Acute Respiratory Syndrome Coronavirus 2 (SARS-CoV-2), individual and public changes to the infection were drastic, sudden, and disruptive in regard to the social norms of society [1]. All of these efforts were executed to prevent the transmission of a deadly pathology, coronavirus infectious disease 2019 (COVID-19) [2]. With the declaration of COVID-19 as a pandemic by the World Health Organization (WHO) in March 2020, many elements of society and community were faced with attempting to assimilate the public health recommendations for infectious control. This assimilation was executed simultaneously by the government, businesses, service agencies, and community organizations in an attempt to continue to deliver key services for a functioning community, services such as, but not limited to, education, transportation, food distribution, and legislation. Therefore, all vital services had to balance delivering their modified services while attempting to stay up to date with COVID-19 information and regulations.

For schools and school systems, guidance on how to best adapt to COVID-19 was scarce and inconsistent. The infection control management of COVID-19 in school buildings was challenging, partly because of the variability in COVID-19 symptoms among younger people, making detection difficult with little evidence for the school setting transmission dynamics early in the pandemic [3,4,5]. Further, school officials, teachers, and administrations had to assist with many public health crises as a result of the pandemic, including transmission prevention strategies (e.g., face masks and physical distancing), and/or the recognition that students would have COVID-19-related personal tragedies (e.g., the loss of a loved one). The resulting sequelae included an overall surge in mental health morbidities among school-age youth and adolescents [6,7,8]. Given the additional challenges of stress and other mental health threats imposed on the school systems, an immediate priority for school personnel was to alleviate these concerns by selecting credible sources and receiving efficient and reliable information to address mental health concerns without stigma. 

In May 2020, a multidisciplinary school-based curriculum was launched in an effort to foster an academic and public health–community collaboration for scholastic organizations. The Health Education and Training (HEAT) Corps was intended to achieve several objectives for students: (a) provide scientific information and health literacy regarding the pandemic and updated COVID-19-related prevention strategies, (b) create a comfortable environment to discuss mental health concerns, and (c) create access to COVID-19 clinicians and public health experts in real time. HEAT Corps was initially launched as a just-in-time curriculum [9]. However, over the course of the pandemic, the curricula evolved to serve the needs of international scholastic communities.

In this review, we highlight the process and feasibility of implementing an international COVID-19 school-based initiative over two years of the pandemic (May 2020 to May 2022). Specifically, we discuss the pedagogy of the curriculum, the training of the HEAT Corps instructors, and additional outcomes that fostered the necessary public health–community partnership to mitigate the impact of COVID-19 on diverse populations.

## 2. The Curriculum: COVID-19

### 2.1. Themes and Content

In identifying themes and content to teach school-age children, we consulted with physicians, nurses, school administrators, youth and public health personnel to understand what would resonate with youth and adolescents to (a) improve science literacy, (b) understand how the pandemic came to be, and (c) what role young people play in ending the pandemic. The curriculum was developed to provide Kindergarten (K)–Grade 12 students with insight into the science around the pandemic, and tangible actions they could use to decrease the spread of COVID-19 in their communities. The topics of interest included “the biology of the COVID-19 virus”, “mathematical models”, “physics of face masks”, “the chemistry of hand hygiene”, “vaccines”, “mental health and wellness during the pandemic”, and “myth busters.” Table 1 includes the complete list of the original eight topics and learning objectives. The theme of mental health developed by clinical experts focused on understanding moods of melancholy, anxiety, and stress without stigma or judgment, and how to access school and local resources if they needed support. 

It should be emphasized that the aforementioned topics, especially mental health, were created after the initial implementation of the HEAT Corp curriculum. Teachers and students requested more information on immediate actions they could take to assist in the promotion of health during the pandemic. The pedagogy review by HEAT Corp personnel assured that the lessons could be reaffirmed through hands-on activities that would serve a purpose during the pandemic. In addition, the mental health and wellness module included key actions that all students and teachers could take in an effort to promote mental health services and resources. Such adaptations were due to the ability to receive feedback and requests from both students and teachers alike. 

Additionally, each topic and module was reviewed by researchers and leaders in education to determine developmental and educational appropriateness for school-age youth. The educators also provided guidance on the flexibility of the material based on the unique needs of their classroom (e.g., a fifth-grade science teacher may determine that their class was advanced enough for the middle school-level version). Finally, all of the modules were combined into one curriculum that was emailed to school personnel (administrators and teachers) for a review of comfort with the material and the appropriateness of the educational level for the respective students prior to presentation.

### 2.2. Content Delivery

Each 45-min module was presented virtually on a secure platform by a pair of trained HEAT Corps volunteers or faculty. The presentations consisted of a 20-min informational content, 15-min questions and answers (Q & A) session, a 10-min recap, and previews of the next lesson. The classroom teachers were invited to assist with the lesson and to encourage the students to interact. The frequency of presentations (Table 1) was based on a schedule determined by the school (e.g., once a week, two times a week, or three times a week). During the academic year 2021–2022, the eight modules were synthesized into a 45-min session that included a 35-min presentation, a 10-min Q & A session, and a recap of the main points. This compressed module was requested by schools to increase the number of students receiving the most pertinent information in a shorter amount of time and to combat the ongoing challenges posed by misinformation. The management of schedules, operational communication with the schools and volunteers, and marketing efforts were coordinated by dedicated administrative team members.

Finally, for each module, two HEAT Corps instructors were always in attendance. One instructor delivered the lesson, while the other was responsible for reviewing the prior lesson as well as providing the current topic recap. Both instructors participated in answering questions during the Q & A session. If any questions could not be answered, one of the instructors would save the question, review it with the HEAT Corps faculty for an answer, and follow up with the classroom teacher to provide the answer.

### 2.3. Training

The all-volunteer personnel recruited for teaching the HEAT Corps curriculum were pre-professional students representing a spectrum of health careers. The majority of HEAT Corps instructors were undergraduate pre-health students (pre-medicine, pre-nursing, and pre-public health), followed by graduate-level medical, nursing, and public health students. A few physicians and nurses joined in the training sessions. Promotion of the training occurred through intranetworks of the university and health system. 

To become a HEAT Corps instructor, recruited volunteers were required to complete several training steps. First, trainees were instructed to watch model videos of each module. Second, trainees would join 2-h live sessions hosted by the HEAT Corps faculty to debrief and discuss each of the eight lessons. During the live sessions, volunteers reviewed the modules slide by slide and asked questions and/or raised any concerns regarding the COVID-19-related material. Further, the volunteers were taught approaches for answering difficult questions, especially questions rooted in political concerns or misinformation. Above all, the training emphasized professionalism and how to maintain decorum during potentially difficult exchanges. In the second of the two sessions, 90 min were devoted to teaching public speaking skills and techniques. The final 30 min of the second session discussed etiquette for teaching in schools as well as any additional child-protective requirements. Of note, the live sessions were recorded, allowing students to re-watch as needed. 

For trainees who also saw this opportunity as a means to improve their public speaking, the program offered additional support for practice and review. The trainees were invited to participate in live practice sessions, record themselves delivering one of the lessons, or both. Whether live or recorded, the HEAT Corps faculty provided feedback on what the trainee had done well, what they could improve, and any other insight that the faculty believed was vital to becoming proficient at public speaking. 

### 2.4. Town Halls

In addition to the in-classroom lessons, the academic public health–community partnership extended its services to assist school faculty, staff, and families with information related to the COVID-19 pandemic. School communities were invited to participate in virtual town halls facilitated by a moderator and expert panelists knowledgeable about emerging COVID issues. The expert panelists ranged from pulmonary and critical care physicians to mental health clinicians and social workers. The moderator was selected from our HEAT Corps faculty, while the expert panelists were identified by the HEAT Corps team from the community, university, or other highly reliable and nonpolitical organizations. 

All town halls followed a format that included brief introductions of the moderator and panelists, followed by designated, preselected questions for each panelist. Once the preselected questions were answered, the audience was invited to ask questions or provide comments. Audience questions were submitted either on the virtual platform used for the town hall (selected in conjunction with the school system), by text message, or both. Each town hall was 60–90 min long, occurring in the evening hours during the academic year and business hours during the summer. If questions remained at the end of the town hall, the moderator emailed questions to the panelists for opportunities to provide answers. Then, a HEAT Corps staff member forwarded answers to school leadership for distribution.

Each collaborating school was welcomed to either request a town hall or join a town hall with another partnering school. At times, school systems, under the guidance of the superintendent, requested a network of schools to join a town hall. Regardless of the potential size of the audience, the outline of the town hall remained consistent. Further, schools and school systems were able to request more than one town hall on the same topic, but for different audiences (e.g., one for school staff and teachers and another town hall for students and families). Of note, if the town hall was requested in a language other than English, HEAT Corps staff identified moderators and experts who spoke the requested language, even if the selected personnel were not immediate HEAT Corps faculty members. Overall, the town halls were designed to accommodate the communities and their concerns during the COVID-19 pandemic.

### 2.5. Feasibility

The HEAT Corps training and curriculum began as a local response to a community request and need. It evolved into a sustaining, multidisciplinary initiative. During the first two years, HEAT training reached 148 schools, over 300 classrooms, and approximately 13,230 students in the K–12th grades. HEAT has presented in 19 states and the District of Columbia, as well as in six countries (Sudan, Guatemala, Cyprus, Canada, Panama, and India). The team has facilitated workshops and town halls in both English and Spanish. The curriculum has been taught by 167 volunteers. This initiative began as an all-volunteer effort by the faculty and administrators as well as instructors. As HEAT evolved, seed financial support was received from the university’s President’s office, and grants are continually being issued.

## 3. Discussion

In implementing an international COVID-19 school-based initiative over two years of the pandemic, the ability to collaborate with school systems was seen as welcomed and needed. The necessity grew from the concern of many school leaders and personnel attempting to balance “staying up to date with COVID-19 and vaccines guidance” with “managing the mental health consequences” of the staff and students. As such, the HEAT Corps initiative shared the responsibilities of the school systems by providing additional resources (e.g., face masks and vaccine access sites) as requested by the communities. Further, the strategy of building an active relationship with each school system and educational organization has allowed the HEAT Corps model to evolve to meet other public health crises that may impact a school and community; such collaboration is established through the ease of access to HEAT Corps (Figure 1). 

A community-based curriculum focused on public health crises designed for school-age youth and adolescents has been a consistent effort for decades. Public health curricula have been designed to assist in the efforts of prevention and mitigation, of crises impacting schools and their respective communities. These crises have ranged from tobacco and electronic cigarette usage and the promotion of exercise and healthy eating choices to malaria control [10,11,12,13]. While others have designed COVID-19-based information to be used by schools [14], it is not implemented (virtually) live by trained instructors such as ours. The aforementioned curricula are often designed with a health end goal, from the prevention of smoking to weight loss. Similar to those curricula, our curriculum is designed with health outcome endpoints, such as face mask usage, consistent hand washing, and vaccination adherence. However, our curricula are also designed to promote science and health literacy. From understanding “cause-and-effect” to “how face masks work”, these portions of the curricula are meant to allow the students, staff, and parents to gain an understanding of how science is used to answer challenging questions as well as react to them in an effort to prevent morbidity. We believe such insight will also assist in identifying misinformation, which attempts to oversimplify and misleading data to fit narratives not designed for public health well-being [15]. Finally, such a focus in this academic public health–community collaboration is meant to strengthen the public trust in science, scientists and clinicians, and public health officials, especially at a time of a pandemic.

Two years into the implementation of the curriculum, the HEAT Corps is undergoing an evaluation as both an educational tool and a public health effort. Teacher and student insight into the modules and specific content will be important for both the purposes of feedback and assessment of the educational/health literacy gain. As a public health effort, evaluation is warranted to understand if the curriculum influenced specific COVID-19 outcomes, e.g., vaccination rates of a school, COVID-19-related reductions in absenteeism by both staff and students, and compliance with face mask policies. In addition, creating curricula on other public health crisis themes is underway (e.g., youth usage of electronic cigarettes, gun violence, and trauma). Further, understanding how the curriculum impacted HEAT Corps instructors will be valuable, specifically if they were able to gain an appreciation for community engagement as an efficient tool for the dissemination of public health information. These directions for evaluation have been gleaned from anecdotal and qualitative feedback. Overall, the HEAT Corps curricula are undergoing extensive evaluation to understand future feasibility and impact, and thereby position this public health intervention for further present and future collaborations with communities and organizations.

## 4. Conclusions

Academic health–community bi-directional collaborations are warranted now more than ever as we face one public health crisis after another. For a community partnering with an academic health system, the gain will be for the efficient dissemination of public health information, with the appropriate pedagogy and andragogy to assure the community will understand the aforementioned information. For the academic health system, the gain is in the ability to promote health and wellness and prevent disease on a cost-effective, larger scale. The ability to partner with a school system assures that such gains are felt across all age groups and diverse populations. The HEAT Corps has created a feasible model for academic health–community collaborations. Moving forward, evaluating the impact of such an initiative will be vital, for both community and health systems, and in doing so, begin to establish the standards needed for sustainable and influential academic public health–community partnerships.

## Figures and Tables

**Figure 1 ijerph-20-01829-f001:**
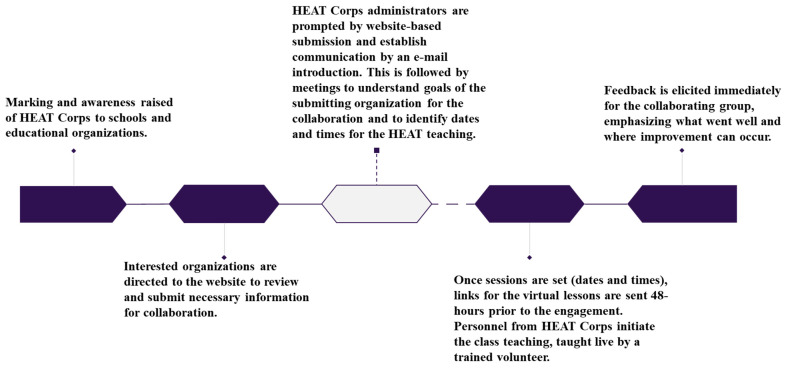
Summary of steps for accessing HEAT Corp information to establish a collaboration with a school and/or educational organization.

**Table 1 ijerph-20-01829-t001:** List of COVID-19 Curriculum Themes.

Title	Objectives:
Biology of the COVID-19 Virus Module 1	Reviewed SARS-CoV-2, specifically key elements that allowed the virus to become a pandemic. Key Terms: zoonotic, incubation period
Mathematical Models Module 2	Discussed the variables that are needed to create a pandemic. Key Terms: Reproduction number, infection
Physics of Face Masks Module 3	Provided students with insight as to how face masks can prevent infection and transmission. Key Terms: Particulate matter, aerosol
Chemistry of Hand Hygiene Module 4	Reviewed how a virus (and other microbes) are impacted by hand washing and hand sanitizers. Key Terms: Oil, viral shell
Vaccines Module 5	Reviewed how vaccines are created, and how the COVID-19 vaccine arrived in time to manage the pandemic. Key Terms: Vaccines, infection, herd immunity
Mental Health and Wellness during a Pandemic I Module 6	Provided an understanding of mental health challenges during the pandemic faced by youth and adults. This covered stress, anxiety, depression, the significance of support systems, destigmatizing mental health, and provided resources for further guidance. This module is divided into two lessons. Key Terms: Mental health, feelings, stress
Mental Health and Wellness during a Pandemic II Module 7	Provided an understanding of mental health challenges during the pandemic faced by youth and adults. This covered stress, anxiety, depression, the significance of support systems, and destigmatizing mental health, and provided resources for further guidance. This module is divided into two lessons. Key Terms: Mental health, feelings, stress
Myth Busters Module 8	Assisted in helping students understand causality and how to better identify misinformation while learning how science evolves during infectious crises. Key Terms: Cause-and-effect, science literacy, alternative theories

## Data Availability

All data can be requested through the corresponding author.

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
