# Peer review of "The Health Education and Training (HEAT) Corps: A Medical–Community Collaboration Response during the COVID-19 Pandemic"

_ijerph, 2023, doi:10.3390/ijerph20031829_

Round 1
Reviewer 1 Report
The authors describe the implementation of a health education and training program, specifically as a response to the COVID-19 pandemic.
My comments are:
Mention if there are programs similar to yours and if so, what makes it different?
Since the virus will already be part of the seasonal infectious diseases in society, what other topics do you plan to include in the short term?
What feedback did you receive during the application of the program?
In which countries was the program applied and what were the criteria for selecting them?
What are the necessary resources and criteria to access your program?
It is suggested to include a diagram or figure showing the procedure to access your program.
Author Response
Reviewer #1:
Mention if there are programs similar to yours and if so, what makes it different?
We did find another program similar to ours, which provided on-line material to use for schools. We do mention our main difference, in that we were live for all schools (virtually) and created a platform where the material would be updated frequently. We mention them on page 11 of our revised manuscript.
Since the virus will already be part of the seasonal infectious diseases in society, what other topics do you plan to include in the short term?
We have current public health concerns being taught such as youth usage of electronic cigarettes and gun violence (page 11).
What feedback did you receive during the application of the program?
Much and this feedback will be presented in our evaluation of the program. Feedback included more hands-on activities to more requests for town-halls. With this current manuscript submission, we wanted to focus on the process and feasibility of implementing the curriculum, setting the stage for several research projects that have come from our work.
In which countries was the program applied and what were the criteria for selecting them?
No criteria to select them. They included Sudan, Guatemala, Canada, Panama, India and Cyprus.
What are the necessary resources and criteria to access your program?
We believe this is captured in Figure 1 (see below). Simply put: it was access to our website.
It is suggested to include a diagram or figure showing the procedure to access your program.
We have added a figure to the manuscript. This is a great recommendation!

Reviewer 2 Report
This paper is presented as a project report and not as a research paper. I shall try to assess its publication value based on what one can learn from the project than on its scientific contribution. To be of value, the learning from the project must be replicable, generalizable, and extensible. While I applaud the authors for the implementation of the project, the learning one can glean from the description as provided in the paper is minimal. That can be corrected for the authors have rich (even if anecdotal) data about what they have learnt, the learning that took place over the two years, and while it was expanded internationally. Towards the end of the paper the authors state the following:
“Overall, the HEAT Corps curricula are undergoing extensive evaluation to understand future feasibility and impact, and thereby position this public health intervention for further present and future collaborations with communities and organizations.”
“Moving forward, evaluating the impact of such an initiative will be vital, for both community and health systems. And in doing so, begin to establish the standards needed for sustainable and influential academic public health-community partnerships.”
I agree about the importance of these evaluations. Perhaps the publication of the paper can wait till the completion of the evaluations. Without them the paper remains a detailed description without feedback and learning.
In the introduction to the paper the authors state the following:
“The Health Education and Training (HEAT) Corps was intended to achieve several objectives for students: a) provide scientific information and health literacy regarding the pandemic and updated COVID-19-related prevention strategies, b) create a comfortable environment to discuss mental health concerns, and c) create access to COVID-19 clinicians and public health experts in real time. HEAT Corps was initially launched as a just-in-time curriculum. However, over the course of the pandemic, the curricula evolved in an effort to serve the needs of international scholastic communities.”
The above are non-trivial, controversial objectives within the USA and abroad. Even as they await the formal summative evaluation, could the authors provide formative (even if anecdotal as cases) evaluation of these objectives. For example, regarding each of the objectives:
· What was the feedback from the students and other stakeholders?
· What was the response to the feedback from HEAT?
· How did the curriculum and pedagogy evolve in response to the feedback and learning?
Author Response
This paper is presented as a project report and not as a research paper. I shall try to assess its publication value based on what one can learn from the project than on its scientific contribution. To be of value, the learning from the project must be replicable, generalizable, and extensible. While I applaud the authors for the implementation of the project, the learning one can glean from the description as provided in the paper is minimal. That can be corrected for the authors have rich (even if anecdotal) data about what they have learnt, the learning that took place over the two years, and while it was expanded internationally.
Towards the end of the paper the authors state the following:
“Overall, the HEAT Corps curricula are undergoing extensive evaluation to understand future feasibility and impact, and thereby position this public health intervention for further present and future collaborations with communities and organizations.”
“Moving forward, evaluating the impact of such an initiative will be vital, for both community and health systems. And in doing so, begin to establish the standards needed for sustainable and influential academic public health-community partnerships.”
I agree about the importance of these evaluations. Perhaps the publication of the paper can wait till the completion of the evaluations. Without them the paper remains a detailed description without feedback and learning.
This is a great point. The goal of this is to serve as a baseline for how we created and implemented the curriculum. It has been a request by many others and we thought providing it as an academic manuscript would be the best way to disseminate our process mechanisms. We believe there is value in knowing how we created and implemented the HEAT project and the feasibility of it.
Agreed that outcomes will be important, and these will be more comprehensive for our next publications. Again, great point for future work.
In the introduction to the paper the authors state the following:
“The Health Education and Training (HEAT) Corps was intended to achieve several objectives for students: a) provide scientific information and health literacy regarding the pandemic and updated COVID-19-related prevention strategies, b) create a comfortable environment to discuss mental health concerns, and c) create access to COVID-19 clinicians and public health experts in real time. HEAT Corps was initially launched as a just-in-time curriculum. However, over the course of the pandemic, the curricula evolved in an effort to serve the needs of international scholastic communities.”
The above are non-trivial, controversial objectives within the USA and abroad. Even as they await the formal summative evaluation, could the authors provide formative (even if anecdotal as cases) evaluation of these objectives. For example, regarding each of the objectives:
- What was the feedback from the students and other stakeholders?
- What was the response to the feedback from HEAT?
- How did the curriculum and pedagogy evolve in response to the feedback and learning?
Great questions. We have provided a general input on feedback, much of it centering on “what can we do to help end the pandemic?” And “what can we do about mental health?” Our HEAT team responded by assuring the immediate taught lessons adapted to actions that local communities can take. The pedagogy evolved immediately to assure hands-on insight and activities occurred with each lesson so that students, teachers, and parents understood what immediate actions they can take. We have revised the manuscript to reflect such input/feedback.

Round 2
Reviewer 2 Report
The authors have addressed my concerns adequately for publication as a report.